



# A NEW AUTOMATED RADIOLARIAN IMAGE ACQUISITION, STACKING, PROCESSING, SEGMENTATION, AND IDENTIFICATION WORKFLOW

Martin Tetard[1], Ross Marchant[1, 2], Giuseppe Cortese[3], Yves Gally[1], Thibault de Garidel-Thoron[1], and Luc Beaufort[1]

[1]Aix Marseille Univ, CNRS, IRD, Coll France, INRAE, CEREGE, Aix-en-Provence, France.
[2]Present address: School of Electrical Engineering and Robotics, Queensland University of Technology, Brisbane, Australia.
[3]GNS Science, Lower Hutt, New Zealand.

**Correspondence:** M. Tetard (tetard@cerege.fr)

**Abstract.** Identification of microfossils is usually done by expert taxonomists and requires time and a significant amount of systematic knowledge developed over many years. These studies require manual identification of numerous specimens in many samples under a microscope, which is very tedious and time consuming. Furthermore, identification may differ between operators, biasing reproducibility. Recent technological advances in image acquisition, processing, and recognition now enable automated procedures for this process, from microscope image acquisition to taxonomic identification.

A new workflow was developed for automated radiolarian image acquisition, stacking, processing, segmentation, and identification. The protocol includes a newly proposed methodology for preparing radiolarian microscopic slides. We mount 8 samples per slide, using a recently developed 3D-printed decanter that enable the random and uniform settling of particles, and minimise the loss of material. Once ready, slides are automatically imaged using a transmitted light microscope. About 4000 specimens per slide (500 per sample) are captured in digital images which include stacking techniques to improve their focus and sharpness. Automated image processing and segmentation is then performed using a custom plugin developed for the *ImageJ* software. Each individual radiolarian image is automatically classified by a convolutional neural network (CNN) trained on a radiolarian database (currently 17,065 images, corresponding to 112 classes) using the software, *ParticleTrieur*.

The trained CNN has an overall accuracy of about 90 %. The whole procedure, including the image acquisition, stacking, processing, segmentation and recognition, is entirely automated via a *LabVIEW* interface, and takes approximately 1 hour per sample. Census data count and classified radiolarian images are then automatically exported and saved. This new workflow paves the way for the analysis of long-term, radiolarian-based palaeoclimatic records from siliceous remains-bearing samples.



## 1 Introduction

The term radiolarians currently refers to the polycystine radiolarian orders Spumellaria and Nassellaria, whose shell is made of opaline silica, relatively well preserved in the fossil record by comparison with the Acantharia and Phaeodaria groups. They are marine micro-organisms whose siliceous shells are found in the sedimentary record since their appearance during the Cambrian period (Boltovskoy, 1999; Lazarus et al., 2015; Suzuki and Not., 2015). While they have been originally neglected for a long time for biostratigraphical studies due to several documented cases of recurrent evolution in the overall morphology of some

taxa (e.g. Schrock and Twenhofel, 1953; Campbell, 1954; Bjørklund and Goll, 1979), radiolarian taxonomy and stratigraphy have significantly progressed due to Deep Sea Drilling Project (DSDP) studies since 1968 (Sanfilippo et al., 1985) and are currently of major interest. Radiolarians are commonly used in biostratigraphy by documenting the presence / absence of key marker species, as well as in palaeoceanographic reconstructions of past productivity, temperature, and variability of water masses, wherein these approaches rely instead on relative species abundances. For both these approaches, radiolarians are

particularly useful in high latitude settings (e.g. the Southern Ocean) where both the preservation and species diversity of calcareous microfossils are very low.

Indeed, radiolarian's delicate siliceous remains have been proved important for decades in micropalaeontological studies focussing on palaeoenvironmental reconstructions from various oceanic areas to investigate primary and export productivity (e.g. Welling et al., 1992; Lazarus, 2002; Abelmann and Nimmergut, 2005; Lazarus et al., 2006; Hernández-Almeida et al.,

2013; Matsuzaki et al., 2019), sea surface temperature (e.g. Abelmann et al., 1999; Lazarus, 2002; Cortese and Abelmann, 2002; Lüer et al., 2008; Panitz et al., 2015; Kamikuri, 2017; Hernández-Almeida et al., 2017; Matsuzaki et al., 2019), water masses (e.g. Welling et al., 1992; Kamikuri et al., 2009; Kamikuri, 2017; Hernández-Almeida et al., 2017; Matsuzaki et al., 2019) and oxygenation (e.g. Matsuzaki et al., 2019) across the Cenozoic. At present, radiolarians assemblages are considered to be consistent and valuable micropalaeontological bio-indicators as they are largely distributed in all oceans since their

appearance and can be very abundant in sediments (e.g. Sanfilippo et al., 1985; Boltovskoy, 1998; Hernández-Almeida et al., 2017).

However, despite their usefulness for such investigations, radiolarians are not as used as other microfossil groups such as benthic and planktic foraminifera, or nannofossils such as coccolithophorids. Experts on living and fossil radiolarians are relatively few, and some radiolarian species still lack a satisfactory taxonomy, especially for taxa within the order Spumellaria

(Riedel, 1967; Sanfilippo et al., 1985). Identification of a substantial and sufficient number of specimens per sample (usually about 300 for reliable assemblage composition estimations Fatela and Taborda, 2002) is very time-consuming and requires a consistent and detailed taxonomic knowledge. Moreover, as it is common for all microfossil groups, and especially true for radiolarians, determination and taxonomy of recovered specimens can be different between studies as it can be biased by the subjective appreciation of the operator, influencing reproducibility of the census counts.

Recent technological advances in image acquisition, processing, and recognition now enable automated procedures, from microscopic slide field-of-view acquisition to taxonomic identification, that can ease radiolarian studies. In the early 1980's, some authors had already proposed to automatically analyse the size and shape of a large number of digitised images of



assemblages of microfossils (Budai et al., 1980), in order to investigate the variability of their morphology and use it as a palaeoenvironmental descriptor. For more than 20 years now, the CEREGE laboratory has been a pioneer in automated

image acquisition and recognition for several microfossil groups. Dollfus and Beaufort (1999) developed a structured multi-layer fat Neural network for coccolith recognition, that was first applied in 2001 to Late Pleistocene primary productivity reconstructions (Beaufort et al., 2001). This formed the base for the following Système de Reconnaissance Automatique de Coccolithes (SYRACO) workflow, that used dynamic neural networks (Beaufort and Dollfus, 2004) and is still operating today. The past few years have seen the emergence and development of convolutional neural networks (CNNs), a deep-learning

approach that enables the automated classification of large sets of images. Several workflows inspired by SYRACO and now using CNNs were successively developed at CEREGE and applied to microfossil taxa (e.g. Marchant et al., submitted; Bourel et al., accepted). For radiolarians, Renaudie et al. (2018) recently achieved promising results focussing on the automated identification of species from the same genus with promising results. They obtain an overall identification accuracy of 73 %, achieved over 16 species from 2 genera, where the morphological difference between species can be very tricky.

In this paper, we also propose a workflow for the automated identification of radiolarians. Our approach differs in that we wanted to generate a neural network that could recognise most of the common radiolarian species, rather than those of a specific genus, in order to investigate their abundance (relative and absolute) and diversity, and thus use them as bio-indicators to reconstruct palaeoenvironmental parameters. It is necessary to obtain a large database of images covering the common species in order to train the network. Out of the modern living 400 to 500 polycystine species, about 100 are relatively common

(Boltovskoy, 1998), however, they have yet to be imaged to create a database for automated recognition purposes. Some online Cenozoic radiolarian databases have already existed for a few years (e.g. WoRaD, Boltovskoy et al., 2010; radiolaria.org, Dolven and Skjerpen, 2006; Radworld, Caulet et al., 2006; see Lazarus et al., 2015 for an extensive review of the existing databases), however these are more directed to creating a catalogue for taxonomic purposes. As such, we created an exhaustive and participative database specifically for CNN training and automated recognition purposes, called AutoRadio (Automated

Radiolarian visible at: https://autoradio.cerege.fr). To achieve this goal, a new protocol to obtain standard images for inclusion in the database was required (square images of individual white specimens on a black background, using a stacking technique if possible) which was also developed in this study.

## 2   Material and methods

### 2.1   Material

Radiolarian microfossils to be used in this study were extracted from several sediment cores. Core MD97-2140 was retrieved from the centre of the West Pacific Warm Pool (WPWP; latitude: 2°02'N; longitude: 141°46'E) at a water depth of 2547 m during the Marion Dufresne IMAGES III-IPHIS cruise in 1997 (Beaufort et al., 1997). This core is currently stored at the CEREGE laboratory, France. The sediments consist of a greyish and compact calcareous nannofossil ooze, also containing abundant radiolarian and foraminiferal faunas (de Garidel-Thoron et al., 2005).



Several samples were chosen to extract siliceous microfossils and thus construct a radiolarian images database. Their depths within the recovered core are: 3-4 cm, (6.3 ka de Garidel-Thoron et al., 2005), 48-49 cm (11.8 ka), 82-83 cm (16.4 ka), 98-99 cm (18.8 ka), 245-246 cm (38.0 ka), 363-364 cm (53.3 ka), 405-406 cm (63.0 ka), 417-418 cm (67.8 ka), 487-488 cm (77.7 ka), 648-650 cm (120.4 ka), 727-728 cm (141.4 ka). For detail on sample processing and slide preparation, the reader is referred to section 2.2.

A few other Miocene to recent samples retrieved from the Warm Pool were also used later to increase the number of rare and absent species in the database. These cores were also taken during the Marion Dufresne IMAGES III-IPHIS cruise: Core MD97-2138 (1760-1761 cm, 2670-2671 cm, 3151-3152 cm); and from IODP Expedition 363 (Rosenthal et al., 2018): Holes U1483A (samples from sections 9H-4W, 14H-5W), U1483B (6H-6W, 18H-2W), U1486B (3H-3W, 6H-4W, 13H-4W), and U1486C (21H-4W).

## 2.2 Random Settling Protocol

A new protocol was developed as a proposed standard methodology for preparing radiolarian microscopic slides. It places 8 samples per standard 76x28 mm slide using 12x12 mm cover slides on which radiolarians are randomly and uniformly decanted using a new 3D-printed decanter (Fig. 1a-b). The 3D file for this new decanter was designed online, using the Autodesk, Inc. 3D design platform *Tinkercad* (https://www.tinkercad.com/), and is available for free at https://github.com/microfossil/Decanter. It was printed on a Raise3D fused deposition layer type printer using 1.75 mm R3D Premium PLA filament for a material cost of about 1 euro. Approximately 30 g of filament was used, and 4.5 hours were needed to print the model using a standard resolution layer height of at least 0.20 mm.

A random settling technique was preferred to a standard smear slide preparation as the objective of this study is a detailed quantitative faunal analysis with investigation of the relative abundances of each taxon (Sanfilippo et al., 1985). Indeed, a random settling technique provides a more uniform distribution of the residue resulting in less clumped particles, which are also easier to capture digital images of each specimen. The new decanter minimises the loss of material, and a slide guide (Fig. 1c) can also be used to align cover slides. During development, various shapes and sizes of tank were tested, and the one presented herein was the best compromise between the quantity of sample material required, loss of residue that would not settle on the slide, and quantity of microfossil residue recovered. This method is an improved version of the original random settling method developed by (Moore, 1973), adapted to radiolarian studies (Boltovskoy, 1998) that provides an even and random distribution of the shells on a slide, and modified by (Beaufort et al., 2014) to mount up to 8 samples on a single micropalaeontological glass slide. The use of this new device is simple: a 12x12 mm cover slide is placed in the middle of each tank and maintained centred by the fins. A solution containing radiolarians in suspension for each sample is then poured onto each tank and after a few minutes of settling on the cover slides, water is vacuumed out from each hole.

The new radiolarian slide preparation protocol follows the following steps (#2 to 7 have been adapted from a similar procedure used to process limestone and calcareous sediments):

1. Weigh the sediment.



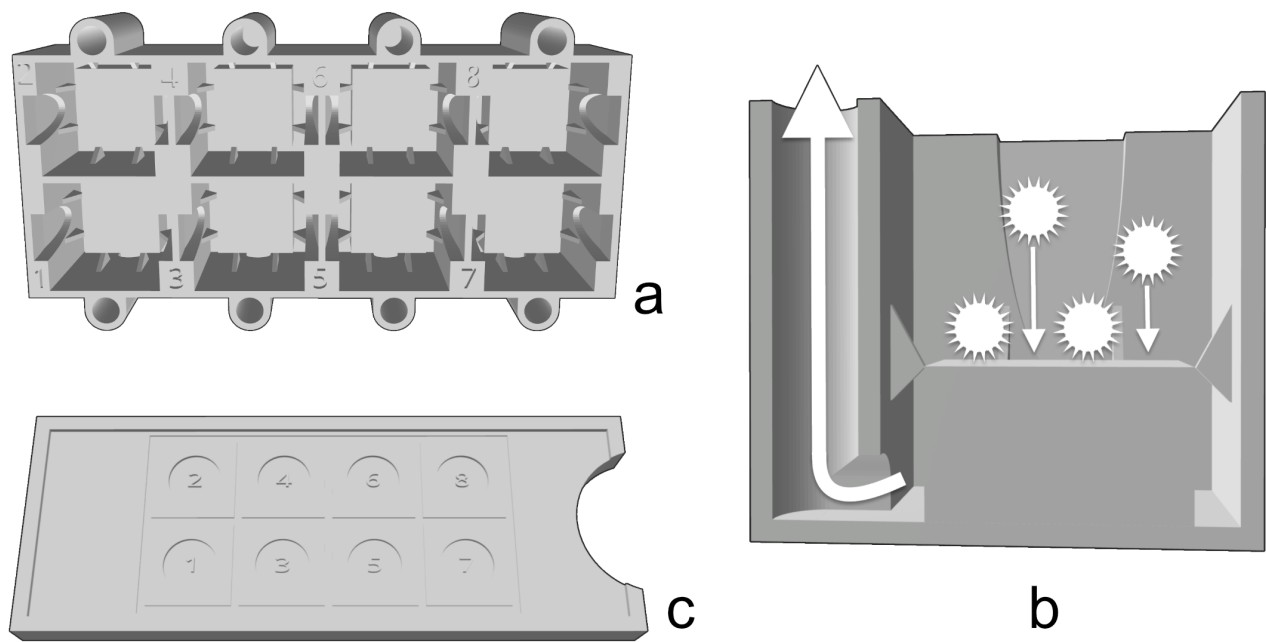

**Figure 1.** a. Upper view of the new 3D-printed decanter, showing 8 tanks. b. Cross-section of a single tank of the new 3D-printed decanter. c. Upper view of the slide guide.

2. Put about 1 g of sediment (depending on the abundance of radiolarians) in a 200 mL beaker and add a few drops of distilled water to disaggregate it.

3. Add a few mL of 37 % Hydrochloric acid (HCl) until the end of the effervescence.

4. Further add a few mL of 10 - 15 % HCl to ensure the end of the effervescence.

5. Pour the solution and rinse the beaker over a 50 $\mu$m sieve.

6. Clean the residues in the sieve until they appear whitish.

7. Rinse the residues using distilled water.

8. Weigh a clean glass storage vial.

9. Pour the residues from the sieve to the vial using ethanol.

10. Once the residues have decanted, remove the excess water using a pipette.

11. Place the vial into the oven (about 50 °C) until the residues are dry.



12. Weigh again the vial to calculate the weight of the recovered residues.

13. Gently tap the vial to unstick the residues from the bottom of the vial.

14. Put a 12x12 mm licked or flame burned cover slide into one tank of the decanter.

15. Take about 0.6 mg of siliceous residue and drop it onto 3.5 mL of distilled water.

16. Shake this solution to suspend the residue and quickly pour it into the corresponding tank.

17. Wait until the residues have decanted (few seconds to minutes) and slowly vacuum out the water from the hole (Fig. 1b).

18. Place the decanter in the oven (about 50 °C) to dry the cover slide.

19. When dry, remove the cover slide from the tank using plastic tweezers and glue it to a standard glass slide (76x28 mm) using optical glue (e.g., NOA81, refractive index of 1.56).

Regarding step #2, the reader should take into consideration the fact that the absolute abundance of radiolarians varies massively in sediment samples from various parts of the ocean. The amount of sediment dissolved into HCl should thus be
customised according to the expected abundance.

Regarding step #15, 0.6 mg of residue corresponds to the best compromise between having a sufficient number of radiolarian specimens and them touching or overlapping too much, for a 12x12 mm cover slide (see Appendix A). This is desirable as often touching specimens cannot be individually segmented from images, leading to "double" images containing two or more specimens, which cannot be easily classified or assigned to a species count. Distilled water was preferred to ethanol as it leads
to less clustering of specimens.

The volume above the cover slide in the tank corresponds to about 45 % of the total volume of the tank. According to the average weight of a radiolarian specimen (about 0.5 $\mu$g Takahashi and Honjo, 1983), 0.6 mg of siliceous residues after chemical treatment should then contain about 1,200 radiolarians, if not "contaminated" by other siliceous particles, of which about 600 should fall on the cover slide, thus resulting in at least 300 specimens that should be available for identification (minimum
required to characterize an assemblage by most of the statistical studies, e.g. Fatela and Taborda, 2002). This was confirmed by our tests showing an average of 473 complete identifiable radiolarian specimens per sample, or at least exhibiting more than 50 % of their shell, including at least the medullary shells for spumellarians, and cephalis and thorax for nassellarians (excluding specimens touching each other and broken specimens). Other testing found that Norland Optical Adhesives NOA81 glue was preferred to other mounting media such as NOA74 or Naphrax due to its refractive index, consistency and long-term
preservation. Although time consuming, metal coating (using C or Au/Pd for example) is also a very efficient way of increasing contrast prior to mounting specimens on the slides. The darkfield illumination technique was too inconsistent in the produced images that further tests were not carried out.



## 2.3 Automated Image acquisition

Particular emphasis was placed on acquiring high-quality slide images, as being able to recognise different radiolarian species
depends on having clearly visible features. For each radiolarian microscopic slide, the 8 cover slides (corresponding to 8
samples) are automatically and consecutively imaged using a Leica DMR 6000 B automated transmitted light microscope
(200x magnification using a HCX PL FLUOTAR 20 x magnification Leica lens) and a Hamamatsu ORCA Flash4.0 LT camera,
controlled via a *LabVIEW* (National Instruments) interface. The microscope parameters were set as: Intensity: 10; Depth of
field: 38; Aperture: 33; and the condenser was lowered by 9 mm from the glass slide. The *LabVIEW* acquisition software
parameters were set as: Exposure: 9 ms; Gain: 1. These settings provided the maximum contrast between the glass shells and
their mounting medium.

For each sample, 324 fields of view (18 x 18 FOVs of 660 x 660 $\mu$m each within each 12x12 mm cover glass) were imaged
using a multi-focal technique (Fig. 2). For each FOV, 15 images were acquired by incrementally stepping the Z focus position
through the microscopic slide (step size: 10 $\mu$m) to cover a total focal distance of 150 $\mu$m, which corresponds to the thickness
of most radiolarian species. This acquisition step takes exactly 1 hour per sample, and thus 8 hours per slide.

## 2.4 Automated Image Processing and Segmentation

Image processing and segmentation is performed via a second *LabVIEW* interface. For each FOV, the batch of 15 images is
automatically stacked using *Helicon Focus 7* (Helicon Soft) and saved following a CoreName-SampleName-FOVNumber.jpg
pattern (Fig. 2). Every stacked FOV image is then processed and segmented into individual specimen images using a custom
plugin (AutoRadio_Segmenter.ijm) developed for the *ImageJ / Fiji* software (V1.52n Schneider et al., 2012). The processing
steps are:

1. Open a stacked FOV image.

2. Subtract its background.

3. Adjust the minimum and maximum greyscale value to increase its contrast.

4. Invert the image and create a mask.

5. Threshold it in order to binarise it.

6. Blur it and threshold it again to obtain the overall shape of each particle.

7. Separate particles that are in contact with each other (require the configurable Biovoxxel "Water Irregular Features"
   plugin, available at: https://github.com/biovoxxel/BioVoxxel_Toolbox).

8. Define regions of interest (ROIs) for each particle.

9. Restore ROIs corresponding to every particle on the original FOV image.





10. Create a square vignette for each particle.

11. Save it into the corresponding "Core" folder and "Sample" subfolder.

Each sample results in approximately 1,000 to 3,000 individual segmented vignettes after the automated image processing
and segmentation step.

## 2.5   Database building and CNN training

*ParticleTrieur* is a dedicated software program developed at CEREGE (Marchant et al., submitted), that enables the operator
to visualise and assign vignettes to manually defined classes, and uses the k-NN (k-nearest neighbours) algorithm to aid
in identification by self-learning and progressively suggesting identification once enough radiolarian pictures are identified.
Using this software, a large dataset of radiolarian taxa images (called the AutoRadio Database) was progressively built (the
original version of the database used in this study can be downloaded at: http://microautomate.cerege.fr/dat). It is currently
composed of 17,065 images, corresponding to 112 classes/taxa. Each class contains between 1 and about 1,000 images.

Once labelled, this database was used to train a CNN (convolutional neural network) for the automated taxonomical identifi-
cation of radiolarian vignettes resulting from the automated microscope image acquisition, processing, and segmentation steps
(Fig. 7). The best results were obtained using a resnet50 topology with added cyclic and gain layers (resnet50_cyclic_gain_tl),
greyscale images resized to 256x256 px, a batch size (number of images presented per training iteration) of 64, 30 epochs
and four drops for the adaptive learning rate system (ALR), and augmentation (Marchant et al., submitted). This training
process lasts about 30 min and generates two files that can then be used for automated recognition (network_info.xml and
frozen_model.pb files).

## 2.6   Automated Taxonomic Identification

Once individual vignettes of radiolarian specimens are generated and saved during the *ImageJ* processing and segmentation
step, they are automatically opened in *ParticleTrieur* using its server mode, controlled by the second *LabVIEW* interface.
These vignettes are then automatically assigned to a class using the trained CNN. Individual vignettes are then automatically
moved into folders corresponding to their core and sample, and subfolders corresponding to their assigned class. Using one
microscope, about 8,000 individuals from two slides (16 cover slides corresponding to 16 samples) can be imaged per day
(about 500 specimens per sample). This fully automated stacking, processing, segmentation and identification step takes about
50 min per sample and operates in parallel to the image acquisition step.

Two types of data are then automatically exported (Fig. 2):

(1) For each sample, a "sample results" file is generated which assembles metadata and morphometric measurements. Each
taxonomic ID is then returned to the *LabVIEW* interface and indexed with its corresponding vignette name (also containing
the core, sample, FOV and vignette numbers in column) into a .txt file for each vignette (in row). For each specimen, morpho-
metric measurements, such as "Area", "Diameter", "Major Axis", "Minor Axis", "Circularity", "Roundness", "Solidity", and
"Eccentricity" are also automatically appended to the .txt file.





**Figure 2.** Automated radiolarian image acquisition, processing and identification workflow. 1, 2 (red rectangle): Automated acquisition steps. 3 (Orange rectangle): Automated FOV images stacking step. 4, 5, 6 (Purple rectangle): Automated FOV images processing and segmentation steps. 7 (Blue rectangle): Automated recognition step. 8 (Green rectangle): Automated export of classified images, census counts, and morphometric measurements.

(2) For each core, census data counts of each sample are automatically compiled. A "core results" file is generated during this process where the abundance of each taxon (in column) for each sample (in row) is automatically incremented.



## 3   Results and discussion

### 3.1   Description of the database

Of the 17,065 images used to construct the database, 112 morphoclasses were created. Of all these classes, 104 belong to Neogene to Quaternary radiolarian species or groups of species (94 classes corresponding to species, 9 to genera, 1 to fam-

ily, see Table 1) and are part of the Spumellaria families Actinommidae, Coccodiscidae, Heliodiscidae, Litheliidae, Pyloniidae, Spongodiscidae, and Tholoniidae; and of the Nassellaria families Artostrobiidae, Cannobotryidae, Collozoidae, Carpocaniidae, Plagiacanthidae, Pterocorythidae, Theoperidae, and Trissocyclidae (see Fig. 3, which includes some example images). Eight non-radiolarian classes (corresponding to "background", "broken" specimens, air "bubble", "diatom", "double", "porous frag-ments", siliceous "particles", and "spicule") were also defined to train the network to recognise these non-radiolarian images

that usually represent 1/2 to 4/5 of the total vignettes.

An extensive overview of the existing Neogene to Quaternary literature was used for the taxonomy and identification of each class, and to define as accurately as possible our assemblages and the observed taxa (including Ling and Anikouchine, 1967; Nigrini and Moore, 1979; Nigrini and Lombari, 1984; Boltovskoy and Jankilevich, 1985; Caulet and Nigrini, 1988; Taka-hashi, 1991; Abelmann, 1992; Boltovskoy, 1998, 1999; Sharma et al., 1999; Nigrini and Sanfilippo, 2001; Itaki et al., 2003;

Lazarus et al., 2005; Kamikuri et al., 2009; Zhang et al., 2009; Matsuzaki et al., 2015; Motoyama et al., 2016; Boltovskoy et al., 2017; Matsuoka, 2017; Zhang and Suzuki, 2017; Sandoval, 2018). Synonymies were also taken into account, especially regarding the work of Boltovskoy (1998, 1999). This means that a few species were regrouped into a single class when a signif-icant morphological gradation was observed and when the limit between the considered species was blurry (e.g., *Eucyrtidium acuminatum* and *E. hexagonatum*; *Sithocampe arachnea* and *S. lineata*; *Actinomma henningsmoeni* and *A. leptodermum*). All

manual taxonomic IDs during the building of the database were reviewed by a radiolarian taxonomy expert (G. Cortese) to ensure consistent and accurate identifications.

### 3.2   Results of the CNN training

One of the best ways to assess the efficiency of a trained CNN is to look at its confusion matrix (Fig. 4). Right before the training step, the dataset is automatically split into two subsets: one being the training set, and the second one the test set.

The data split chosen for this study is 1/5. This means that 4/5 of the original images are used for training (training set) while the remaining 1/5 (test set) of the original images are used for testing the CNN efficiency by calculating several indices. The efficiency results are then represented by the overall accuracy Eq. (1), precision Eq. (2) recall Eq. (3), and individual recall for each class, with these terms defined as:

$$Accuracy = \frac{Number\ of\ images\ correctly\ classified}{total\ number\ of\ images} \tag{1}$$





**Figure 3.** Examples of radiolarian vignettes generated by the automated acquisition, processing and recognition workflow. (a) *Lamprocyclas maritalis*. (b) *Lamprocyrtis hannai*. (c) *Theocorythium trachelium*. (d) *Pterocanium trilobum*. (e) *Pterocanium praetextum*. (f) *Eucecryphalus sestrodiscus*. (g) *Eucyrtidium acuminatum / hexagonatum*. (h) *Acrosphaera spinosa*. (i) *Solenosphaera chierchiae*. (j) *Collosphaera tuberosa*. (k) *Didymocyrtis tetrathalamus tetrathalamus*. (l) *Hexacontium* spp. (m) *Stylatractus neptunus*. (n) *Heliodiscus asteriscus*. (o) *Tetrapyle octacantha* group. Scale bar 100 $\mu$m.

It is the overall performance of the system regardless of class. If you select a random image from the dataset and classify it, the overall accuracy is the probability (in %) that the returned classification is correct.

$$Precision = \frac{Number\ of\ images\ that\ were\ classified\ as\ class\ N\ and\ actually\ belong\ to\ class\ N}{total\ number\ of\ images\ classified\ as\ class\ N} \tag{2}$$





Precision is a metric for a specific class: it is the probability (in %) that an image classified as class N is actually from class N, divided by the total number of images classified as class N.

$$Recall = \frac{Number\ of\ images\ in\ class\ N\ that\ were\ correctly\ classified}{total\ number\ of\ images\ in\ class\ N} \qquad (3)$$

Recall is, for a specific class, the probability (in %) that a random image from class N is correctly classified, divided by the number of images belonging to class N. Recall is basically the accuracy of a single class. Individual recall scores for each class are visible in the confusion matrix (Fig. 4) as the % of class N (in row) that was identified as various classes (in column). For example, for the first row "*Acanthodesmia vinculata*", 73 % of the images belonging to this class were correctly identified,

while 18 % were classified as "*Tholospyris* sp" and 9 % as "broken". If the CNN training was perfect, the diagonal should only exhibit "100" values. The single overall recall and precision scores are the respective values averaged across all the classes.

During the CNN training, all classes containing less than 10 images (corresponding to rare species, currently lacking images) were automatically fused into a single "other" class. Of the original 112 classes, 84 classes (including 76 radiolarian classes) were then trained to be recognised with a current overall precision of about 90 % (89.6 %) over every class. The average

precision is above 83 % (83.1 %) and the average recall is above 80 % (80.2 %). A closer look at the matrix shows that classes with a low recall score usually correspond to classes containing an insufficient number of images (rare species, difficult to get on slide, that would require a significant amount of samples to be processed before enough individual images are generated), usually less than 30 images (e.g., *Collosphaera tuberosa*: 33 %, contains only 16 images; *Dictyophimus gracilipes*: 40 %, contains 26 images), while about 300 images per class are usually recommended.

An investigation of the individual recall for each class (Fig. 5) shows that a minimum recall of 80 % is achieved when the test set for this class (19 out of 84 trained classes; Fig. 5 green square) contains at least 30 images (30 images in the test set = 1/5 of the original set) means that 120 images are contained in the training set (4/5) and that 150 images represent the original dataset for this class; Fig. 5 green square). When the test set is below 30 images per class, a higher number of classes (28 out of 84 trained classes; Fig. 5 orange square) show a recall score below 80 % (59 % in average). More images, about at least

150 (ideally 300) in total for each class, as defined above, are then likely to increase the recall and accuracy of these under-represented classes up to at least 80 %. To this aim, the database will be updated and populated gradually trough the automated processing of new samples. As the aim of this database is to be free-access and participative online, people are encouraged to send and / or add pictures of these under-represented classes.

Surprisingly, a significant number (36 out of 84 trained classes; Fig. 5 blue square) of classes does not seem to require this

number of images in their dataset, as less than 30 images in the test set was sufficient to reach a recall of more than 80 % (e.g., *Pterocanium trilobum* and *P. praetextum* showing a recall of 100 % with only 33 and 61 images in their original dataset, respectively, and 7 and 12 images in their test sets; Fig. 5 blue square). Finally, only a single class (*Porodiscus* sp; 1 class out of 84 trained) contained more than 30 images in its test set, and scored just less than 80 % (79 %; Fig. 5, red square).



**Figure 4.** Confusion matrix showing the overall and individual accuracy, precision and recall for the 84 trained classes.



**Figure 5.** Plot of the number of images in the test set for each trained class (logarithmic scale) vs recall score per class.

## 3.3 Accuracy of the trained CNN on a random set of samples

In order to test the reliability and reproducibility of our trained CNN on actual samples, a slide on which 8 cover slides containing siliceous particles from 8 random samples with variable radiolarian abundances from cores MD97-2138 and MD97-2140 was prepared, and their identification scores computed. This slide was automatically imaged, FOV pictures were automatically segmented and individual vignettes were automatically identified using the trained CNN. After a manual verification of every automated identification, 6 indices were computed: (1) the % of radiolarian images recognised as radiolarians (Fig. 6a); (2) the

% of radiolarian images recognised as the correct radiolarian taxa (Fig. 6c); (3) the % of non-radiolarian images recognised as non-radiolarian particles (Fig. 6b); (4) the % of non-radiolarian images recognised as the correct particle class (Fig. 6d); (5)



the % of non-radiolarian images recognised as radiolarian (non-radiolarian false positive; Fig. 6e); and (6) the % of radiolarian recognised as non-radiolarian (radiolarian false positive; Fig. 6f).

Overall, 10,288 vignettes were identified and manually checked among 8 samples containing between 623 and 2,372 images
each. The abundance of radiolarians ranges from 41 to 340 specimens per sample. The results show that the 6 indices exhibit very close values between the 8 samples. In average, the proportion of radiolarians actually recognised as radiolarian is very high, about 98 % (Fig. 6a) and the proportion of radiolarians identified as the correct radiolarian taxa is about 90 % (Fig. 6b). Almost all radiolarian images are thus recognised as radiolarian with a 10 % error regarding their species identification. Regarding the non-radiolarian images, more than 99 % are recognised as non-radiolarian (Fig. 6c) and about 98 % are assigned
to the correct class (Fig. 6d).

False positive identifications were also investigated and are relatively low. Among all the images identified as non-radiolarians, only 0.34 % should be assigned to radiolarians, and among all the images automatically recognised as radiolarians, about 4 % are non-radiolarian images. Within these 4 %, most of the non-radiolarian images confused with radiolarians exhibit radiolarian features and correspond to the non-radiolarian classes "broken" and "double" that either contain incomplete radiolarians, or
radiolarians touching each other and cannot be assigned to a single species. These false positives are then usually assigned, in the "broken" class case, to the species partially present in the image, or in the "double" class case, to one of the species that can be distinguished.

### 3.4 Biostratigraphy

To explore the applicability of the automated radiolarian identification workflow for biostratigraphic studies, radiolarian faunal
events, such as first occurrences (FOs) and last occurrences (LOs) of radiolarian taxa (about 30 zones were defined for the Cenozoic (Sanfilippo et al., 1985)) were compiled into an Excel spreadsheet. Here, we decided to focus on the biostratigraphy of the Neogene to Quaternary interval using the existing zonation (Nigrini, 1971; Lazarus et al., 1985; Johnson et al., 1989; Moore, 1995; Sanfilippo and Nigrini, 1998; Nigrini and Sanfilippo, 2001; Vigour and Lazarus, 2002; Nigrini et al., 2005; Sanfilippo et al., 1985; Kamikuri, 2017) and especially the recent work of Kamikuri et al. (2009) who compiled and documented the
stratigraphic occurrences of 115 Neogene and Quaternary radiolarian species recovered from Ocean Drilling Program (ODP) Sites 845 and 1241 in the tropical Pacific Ocean.

The known stratigraphic ranges of species included in our database were then compiled into an Excel spreadsheet that automatically suggests the age of any sample, according to the composition of its radiolarian assemblage. As this spreadsheet follows the architecture of the automatically generated census data file for each core (core results file), it can be easily filled by
copy-pasting this content. This operative workflow, that is automated from the image acquisition to the census counts and can suggest an age for the processed sample could thus contribute to the field of biostratigraphy.

### 3.5 Application to other datasets and other studies

To test the potential application and limits of our trained CNN on existing sets of images, we compiled various individual images of radiolarians from the literature including unstacked optical microscope images and scanning electron microscope







**Figure 6.** Identification indices evaluated on 8 random samples recovered from cores MD97-2138 and MD97-2140.

(SEM) images. We then performed a simple colour inversion of the optical microscope image to obtain white specimens on a dark background. Of the hundred images tested, about half were correctly recognised while the other ones were mostly



assigned to the "background" class, likely due to the blurry shell edges of the unstacked images, and to the "broken" class, as only part of the shell were probably recognised. While this 50 % accuracy on a random set of unstacked optical microscope and SEM images may seem relatively low and arbitrary, it is very encouraging and promising for the development of future
and extensive neural networks for automated radiolarian recognition regardless of the imaging method.

## 4   Conclusions

A new automated radiolarian workflow was developed and consists of a sequence of six steps:

1: A new microscopic slide preparation protocol to enable an efficient automated image acquisition on transmitted light microscopes and decrease the loss of material, as this can limit the investigation of samples where radiolarians are scarce.

2: Automated microscope image acquisition that can automatically image microscopic slides bearing up to 8 samples (324 FOV images per sample) at different focal depths (15 images per FOV, every 10 $\mu$m in depth).

3: Automated stacking of each batch of FOV images (using depth maps) to generate a single clear FOV with clearly distinct radiolarian specimens.

4: Automated FOV image processing (contrast enhancement, B&W inversion) and segmentation to generate individual
images for every radiolarian specimen.

5: Automated radiolarian recognition using a CNN, as well as calculating morphometric measurements.

6: Automated export of census data per sample (usually about 500 radiolarian images per sample), and storage of radiolarian images in folders corresponding to their taxonomic identification for every sample.

The whole procedure is then entirely automated from the image acquisition to the census counts, and only required from
the operator to prepare the micropalaeontological slides and put them under the microscope. The operative workflow described in this study can thus perform complex, tedious, time-consuming tasks such as taxonomic identification and census counts by producing reliable, reproducible, and accurate results. This workflow is achieved using a polyvalent and extensive radiolarian image database (currently 17,065 images) and a ResNet CNN trained using transfer learning for modern and Neogene radiolarian identification. The CNN is currently able to recognise 84 classes with an average precision of about 90 %, an overall score
that was also obtained on a test performed on 8 random samples containing more than 10,000 images. In order to continue to increase its efficiency, more images are required, particularly so for rare species. To this aim, the database was made free-access and participative to increase the number of images, especially for rare species where the recall score is relatively low, most likely due to low numbers of training images for these taxa.

This new workflow and associated CNN has the potential to make paleoclimate studies more approachable and feasible,
along with biostratigraphy for very long sequences. The radiolarian census data can then be used to investigate the radiolarian assemblages variability for biostratigraphical purposes, and to develop, apply and improve existing assemblage-based palaeoenvironmental proxies such as SSTs (e.g., radiolarian-based palaeotemperatures for the late Quaternary, Cortese and Abelmann, 2002; Subtropical (ST) Index, Lüer et al., 2008; Radiolarian Temperature Index (RTI), applied to Miocene samples, Kamikuri, 2017; and paleoproductivity (e.g., Upwelling Radiolarian Index (URI), Caulet et al., 1992; Water Depth Ecology



index (WADE), Lazarus et al., 2006). It also enables the investigation of evolutionary trends, appearance of new species, and rate of evolutionary change, a fascinating topic regarding radiolarians and other microfossil groups.

This dataset and following studies also enable the fast and accurate measurement of numerous morphometric parameters for each vignette that was assigned a class in the automated recognition step. In addition to the previous research applications, the morphometry aspect provides the possibility to investigate the link between the morphological variability of a species or an assemblage through time along a sedimentary record and elaborate/test scenarios to explain such variability. This new workflow will now be used on two Neogene to Recent sedimentary records from IODP Expedition 363 (Hole U1483A, Hole U1488A), recovered in the West Pacific Warm Pool.

*Data availability.* The original version of the AutoRadio database used in this study can be downloaded at: http://microautomate.cerege.fr/dat. It is currently composed of 17,065 images, corresponding to 112 classes/taxa.

*Code and data availability.* A manual version of the AutoRadio_Segmenter.ijm plugin (automated image processing performed on *ImageJ / Fiji*), developed to process a root folder ("Core"), containing subfolders ("Samples") of images ("FOVs") is available online for free at https://github.com/microfossil/ImageJ-LabView-Scripts. To use it, download the .ijm file and save it into the ImageJ/plugins folder, and it will be available to use after restarting *ImageJ / Fiji*.

**Appendix A**

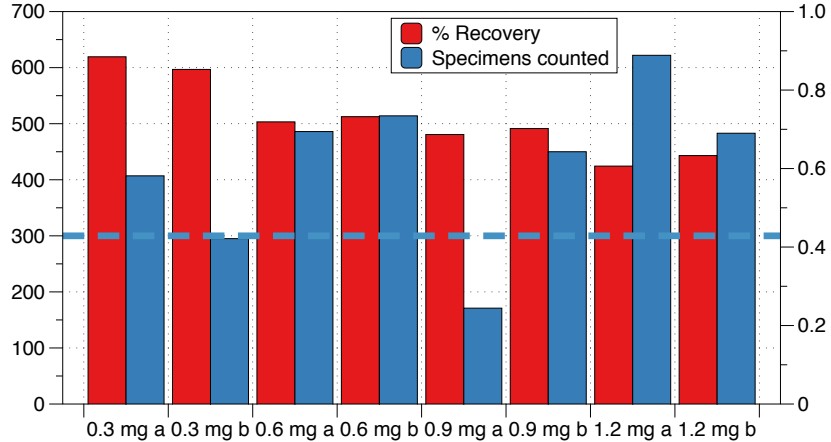

**Figure A1.** Number of specimens counted and % for different weight of radiolarian material drop in the decanter. The dashed blue line corresponds to the minimum counts required per sample.





*Author contributions.* MT designed the experiment, performed its technical aspects, including the image preprocessing and write the first draft of the manuscript. RM developed ParticuleTrieur. MT and GC established the taxonomy of the training set. YG developed the automation of the microscope. LB and TdGT were at the origin of the project. All authors participated in the writing of the manuscript.

*Competing interests.* The authors declare that they have no conflict of interest.

*Disclaimer.* TEXT

*Acknowledgements.* We thanks IODP-France for financial support for this project. This work was also supported by the French National Research Agency (ANR) as part of the French platform called Nano-ID (EQUIPEX project ANR-10-EQPX-39- 01) and the ANR project FIRST (ANR-15-CE4-0006-01). We also thanks the program Ocean Acidification from the french Foundation for Research on Biodiversity (FRB), and the Ministry for the Ecological and Inclusive Transition (MTES) in supporting the project COCCACE.



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
