# Peer review of "A NEW AUTOMATED RADIOLARIAN IMAGE ACQUISITION, STACKING, PROCESSING, SEGMENTATION, AND IDENTIFICATION WORKFLOW"

_Climate of the Past, 2020_

## Referee Comment (RC1) · David Lazarus (Referee) · 21 Jul 2020

General comments

As the authors note, human collection of routine occurrence data for radiolarians or other organisms is time consuming, requires rare, expensive expert workers, and suffers from inconsistencies between data collectors. Automated collection of occurrence data is likely to prove revolutionary to fields such as micropaleontology, where the vast numbers of specimens and species preserved in the fossil record means that the quantity and quality of the data can be expanded by orders of magnitude. This is a rapidly developing subject and papers are appearing in quick succession. Most work in micropaleontology however so far has been on the most intensely studied groups - pollen, planktonic foraminifera and coccolithophores. Only a very few studies have been done on radiolarians, despite their importance to carbon cycling, polar biogeochronology and evolution research. This ms is, so far as I know, the first attempt to essentially throw an AI system at an entire assemblage without a human pre-selection process of the input images. It is a very valuable contribution to see how well this works, and where bottlenecks or barriers arise.

One very important advantage of this as a fully automated system, including primary image acquisition, is that, having identified common categories and trained the system to identify them, future image acquisition can concentrate on the less common categories, resulting in a great improvement int the efficiency of the taxonomic specialist who should be examining and tagging only unknown/poorly documented-rare forms. They might wish to point this out more clearly in the ms.

They post their data and code, or pre-prints very openly - kudos. As Marchant et al. submitted gives the technical component of the AI system I will not comment further here on it.

The ms, and the study itself, are well done and the ms is worthy of publication with only moderate revisions. Nonetheless there are several things that should be improved prior to acceptance.

Critique

Citation

The ms does not cite some other prior attempts to automate radiolarian identification, should try to cite some of these since there are so few of them: e.g. Apostol, L. A., Ma Ìąrquez, E., Gasmen, P., and Solano, G. (2016); Keçeli, A. S., Kaya, A., and Keçeli, S. U. (2017).

Images and Taxonomy

Stacked glassy images are used in this project as the basis for identification. These are arguably the best single image to use if a study is done with just one image per specimen (not actually a requirement for this type of work). However, stacked images often do not show important interior characters (for example in actinommids, pylonids and plagiacanthids, and in many instances in other families as well). These interior characters are important for species identification in many taxa and I found the images sometimes to be frustrating to interpret as an expert in the taxonomy of these forms. The standard method of imaging and presentation in research for this material is to show a manually focussed set of a few unstacked images in transmitted light. Possibly there is a way to use the raw images better? Also, precisely because such stacked images are uncommon in the field it is not clear how easy it will be for community members to add to their open image database, despite the laudable call for contributions. Very few researchers for that matter have access to the, for micropaleontologic standards, complex, expensive equipment used by the authors in this study, tho perhaps only manually generated images are needed as input.

The images are only ca 300 pixels in size. This is a bit marginal. While sufficent for most features, taxonomically important small features can get lost (bladed vs cylindrical spines, etc). Computation costs increase with image size but if possible I would suggest nearly doubling the image resolution for future work, particularly if the database used is going to be promoted as a standard for future contributions.

The cited 17K image database is significantly smaller in numbers of species images. There are non radiolarian broad categories like 'spicule' - 8 categories, ca 8K images, plus supraspecific spp group categories - nearly 20 named,ca 4K images. The number of tagged species images is thus well under 10K. This distinction should be noted clearly in the ms.

For many taxa images have multiple image types, e.g. Acrosphaera spinosa specimens largely covered with some sort of milky bubble (preparation or image stacking artefact?) - 20 of 60 images. This problem is fairly common, seen in many folders.

This particular image problem does not appear in routine sample preparation of similar materials and should be explained, as also the effect on identification accuracy.

There are many, mostly minor issues with the taxonomic image database, tho some of these do not appear in the ms itself due to dropping rare image categories in the analysis.

There are really too many incorrect names being used in the ms and supporting image database. While not a problem for testing (the name tags as such have no effect on the system) these should be corrected. I have not gone through all 100 folders but based on a sampling of these I note the following. 'Strichocorys spp' - no such genus, content appears to be Phormostichoartus pitomorphus, Theocalyptra davisiana - correct name (since 30 years) is Cycladophora davisiana, Calyptra cervus instead of intended? name Corocalyptra cervus, tho recommended name is Eucecryphalus cervus - and see also below, Pseudodictyophimus gracilipes, kilmari, not Dictyophimus (again, since decades) - which importantly changes the family assignment.

There are also some typos such as Stylotractus instead of Stylatractus universus. Lastly there is at least one instance of possible oversplitting - i. e. Dictyocoryne truncatum vs Euchitonia triangulum. What is the difference?

There are also a certain number of specimens in the individual taxon folders that are not con-specific. A brief sampling yields:

image name given correct name total images in category

14875 Cyrtopera langucula Artopilium undulatum 6

11233 Anthocyrtidium ophirense unknown but pores much too large to be conspecific 20

01235 Zygocircus productus Zygocircus piscicaudatus 67

13329 Amphisphaera sp. B appears conspecific w. some specimens labeled as Druppatractus irregularis 21/28

mult. Calyptra cervus multiple Cycladophora species including [u1486-..] Cycladophora cabrilloensis

The large majority of the identifications (so far as the image quality allows) appear to be correct as monospecific classes, even if sometimes the name for the class are not. An attempt to provide a standard set of names (provisional of course as taxonomy is always being revised) was given by a group of taxonomists in Lazarus et al. 2015. I sugggest for the sake of data standardization that they use these, or at least provide a taxonomic appendix where they explain any variant usage.

Performance measurements

The time has come, the Walrus said, To talk of many things: Of shoes and ships and sealing-wax, Of cabbages and kings (Lewis Carroll, Through the Looking Glass, 1871)

Radiolarians are to a highly unusual degree morphologically diverse. There are at least a dozen topologically highly distinct Baupläne alone in living/late Neogene assemblages. Distinctions between these broad morphologic groups are trivially easy. This is quite in contrast to most other clades of organisms, and in particular to planktonic foraminifera, which were used by this research group to initially develop their algorithms and work flows. Planktonic foraminifera are morphologically very conservative (once described to me -by a foram specialist- as 'basically just popcorn'), and can be considered for an imaging system as a single broader category for analysis purposes. For radiolarians, it is really not very informative to know that the system can distinguish between forms as radically different as, well, cabbages and kings. The true test of performance is its capacity to accurately identify and distinguish between species within topologic/taxonomically similar groupings, such as within radiolarian families/subfamilies. This is not only for purely taxonomic reasons, but also for the utitily of a system in applied research. Radiolarians encode ecologic-environmental information almost entirely at the level of individual species. Attempts to use genus

or family level taxa as proxies in applied paleoenvironmental research have yielded almost no useful signals. Geologic age in the Cenozoic is partially recoverable at the genus level, but at a resolution so poor as to make it uninteresting for actual use.

The ms therefore needs to clearly separate the performance of the system in distinguishing between morphologically/taxonomically similar forms and the ability to distinguish extremely dissimilar forms. I suggest in both the statistical analyses of output and in the creation/ organisation of the figures, and the image database for download, that a clustering to higher categories is done, e.g. radiolarian orders and families, plus 'other' for non radiolarian categories such as particles, diatoms or background. The results for pairings of related taxa such as Lophophaena hispida and Peromelissa phalacra, both within the lophophaenid subfamily of Plagiacanthidae (performance values ca 70%) suggest that the accuracy for applied uses may be significantly lower than the current bulk statistics suggest.

It is also important to report separately performance in identifying radiolarian species vs identifying broad categories such as 'spp.', diatoms, particles etc.

Lastly, when errors Are made, the nature of these is significant. It is hard to understand how this system could mis-identify a Pseudodictyophimus gracilipes [incorrectly named Dictyophimus gracilipes in the ms] with Hexacontium spp. - these taxa are in different orders, and have [to a taxonomic expert] fully different morphologies. Some sort of statistic giving not just the error rate, but the type of error - misclassified as to species in same family, different family in order, or different order should be given.

Completeness of taxonomic/morphologic coverage

The authors have clearly made an effort to look at the entire assemblage of radiolarians, which is perhaps the most distinctive, laudable and novel aspect of this study. It must however be noted that the number of actual examined species is less than 80, while radiolarian diversity in tropics-subtropics is ca 500 (just in the sediments, not counting those, relatively few, species in the plankton that do not preserve). Thus only

about 15% of the diversity in these assemblages has been incorporated into the study. Adding a larger percentage of the species is a stated goal of their project and quite correct. However, as more species are added, the number of closely related species pairings will increase, in relation to comparisons between distantly related forms. This is likely to have a negative effect on the performance of the system, as accuracy in similar pairings appears to be fairly low at present. This may not affect using the system for classic assemblage based proxies of paleoenvironmental conditions as these can be based on relatively few, selected species or even species groups. There is not much demand however at the for this type of work, as it has been largely displaced by geochemical methods. Possibly having a cheap system to generate the data will revive it, but I am somewhat skeptical. Biostratigraphy however remains important, along with a variety of emerging themes related to evolution and biodiversity. These studies though need to use a larger fraction of the assemblage, distinguish closely related forms, and/or include rarer species. (Indeed it seems to be an instance of Murphy's Law that important biostratigraphic markers are so hard to find in many slides...). The usability of this system will only become apparent when it has grown to include more taxa, including many closely related and rare forms. The ms should make these limitations clear. There is no general answer to what level of accuracy is 'adequate', but I would suggest that for many biostratigraphic and biodiversity studies error rates should be closer to 1% than 10%, which may prove quite challenging for AI systems to achieve. Alternatively analytic methods in these fields will have to be revised to handle much noisier data.

There is an additional issue in the adequacy of counting only a few hundred individuals to represent an assemblage of radiolarians. This is adequate only for the rather small fraction of species (usually <10% of the species richness) that are present at several percent abundance in the assemblage, as a closer reading of the short paper the authors cite (Fatela and Taborda) would reveal. While a cut-off of several percent is indeed frequently used in paleoenvironmental proxy studies, this is a seriously inadequate degree of coverage for either biostratigraphy or evolution/biodiversity research.

Indeed, this problem is indirectly illustrated in the inadequate numbers of specimens for many species in their training-test image sets. There is in fact a very large body of sophisticated literature in ecology on determining the adequacy of sample sizes for different degrees of assemblage diversity and desired completeness of the resulting sample. See as starting points Chao et al. (2020) Ecol. Res.; Dornelas et al. (2012) Proc. R.Soc. B, and the brief discussion related to radiolarian assemblages in Lazarus et al. (2018) PeerJ. For radiolarians, in many cases the appropriate sample size is several thousand specimens.

Sample coverage

Detailed information is given for this in the SOM, but essentially all material is from a single location in the western equatorial Pacific. I miss a discussion, or at least a disclaimer, of how geographic variation in morphology, or variation over time in lineages might affect the system's performance by blurring between species distinctions. It would also be nice to know in more detail the ages of the samples and the sources of the age information.

New sample preparation method

In this section a new variant of a coverslip holder is described. Although the goal of making slides with very few individuals seems to me to somewhat quixotic given the sample adequacy issues mentioned above, the idea of a custom designed holder that can be manufactured by 3-D printing is novel and is, adapted perhaps to full size cover slips, a useful addition to the literature. The chemical and other preparation steps are fairly standard and, though important to include, could be moved to the SOM.

Figures and Tables

The confusion matrix (fig. 4) is useful but a much more readable table listing each species, numbers/percents correctly and incorrectly classified and the top 3 error categories they were assigned to would be very helpful. I spent too much time scrolling

figure 4 around my screen.

Tiddles

I think the citation to Lazarus et al. 2015 line 23 should be Lazarus 2005 while line 235 should be to Lazarus et al. 2015 - they have inverted these.

―――――――――――――――――――

---

## Author Comment (AC1) · 6 Aug 2020

For each point mentioned by the reviewer, a detailed answer is developed below.

General comments

Comment: As the authors note, human collection of routine occurrence data for radiolarians or other organisms is time consuming, requires rare, expensive expert workers, and suffers from inconsistencies between data collectors. Automated collection of occurrence data is likely to prove revolutionary to fields such as micropaleontology, where the vast numbers of specimens and species preserved in the fossil record means that

the quantity and quality of the data can be expanded by orders of magnitude. This is a rapidly developing subject and papers are appearing in quick succession. Most work in micropaleontology however so far has been on the most intensely studied groups - pollen, planktonic foraminifera and coccolithophores. Only a very few studies have been done on radiolarians, despite their importance to carbon cycling, polar bio-geochronology and evolution research. This ms is, so far as I know, the first attempt to essentially throw an AI system at an entire assemblage without a human pre-selection process of the input images. It is a very valuable contribution to see how well this works, and where bottlenecks or barriers arise.

Response: We thank the reviewer for all the constructive comments about the MS, the aim is to propose, on the long term, a network that would be able to identify most of the common species worldwide, and on longer timescale. Overall, to briefly sum up the corrections made : we improved the training set by adding more than 4000 images, and fine tuning the taxonomy according to the comments of the reviewer, and we trained a new network that uses this new training set.

Comment: One very important advantage of this as a fully automated system, including primary image acquisition, is that, having identified common categories and trained the system to identify them, future image acquisition can concentrate on the less common categories, resulting in a great improvement in the efficiency of the taxonomic specialist who should be examining and tagging only unknown/poorly documented-rare forms. They might wish to point this out more clearly in the ms.

Response: Indeed, focusing on rare species is difficult, as only few specimen can be found in the entire core. To this aim, images of rare species will be progressively added to the network when more samples / cores will be identified. Thank to this system, a taxonomist expert can focus on analyzing rare species, this is added to the conclusions.

Comment: They post their data and code, or pre-prints very openly - kudos. As Marchant et al. submitted gives the technical component of the AI system I will not

comment further here on it. The ms, and the study itself, are well done and the ms is worthy of publication with only moderate revisions. Nonetheless there are several things that should be improved prior to acceptance.

Response: One of our goal that to make as much data and code open so that most people can use it quickly in their lab.

Critique

Citation

Comment: The ms does not cite some other prior attempts to automate radiolarian identification, should try to cite some of these since there are so few of them: e.g. Apostol, L. A., Ma ÌA Ìĺrquez, E., Gasmen, P., and Solano, G. (2016); Keçeli, A. S., Kaya, A., and Keçeli, S. U. (2017).

Response: These missing references were added to the MS lines 63 to 65.

Images and Taxonomy

Comment: Stacked glassy images are used in this project as the basis for identification. These are arguably the best single image to use if a study is done with just one image per specimen (not actually a requirement for this type of work). However, stacked images often do not show important interior characters (for example in actinommids, pylonids and plagiacanthids, and in many instances in other families as well). These interior characters are important for species identification in many taxa and I found the images sometimes to be frustrating to interpret as an expert in the taxonomy of these forms. The standard method of imaging and presentation in research for this material is to show a manually focussed set of a few unstacked images in transmitted light. Possibly there is a way to use the raw images better? Also, precisely because such stacked images are uncommon in the field it is not clear how easy it will be for community members to add to their open image database, despite the laudable call for contributions. Very few researchers for that matter have access to the, for

micropaleontologic standards, complex, expensive equipment used by the authors in this study, tho perhaps only manually generated images are needed as input.

Response: We are aware of the issue that the internal features are usually not visible on our specimens. This is why some grouping of species, or taxa identified at a higher rank (e.g. pylonioid spp) were created. As pointed out by the reviewer, when a single image can be used, as in this study, stacked images are the best choice, for routine and unsupervised analyses. As the stacking occurs directly on the FOV, and not on the vignettes, it is not easy, to directly work on unstacked vignette in our workflow. Moreover, using several images (at least one focused on the centre of the shell, and one on the outside), for the identification of a single specimen is not something we can achieve at the moment. Regarding the use of unstacked image with our database, very encouraging results were produced (see last paragraph of our discussion). To address this issue properly, even if stacking is a more and more used technique in micropalaeontology, we encourage scientist with a large set of unstacked image to share them, in order to integrate them in our network, or create another specific network dedicated to unstacked images.

Comment: The images are only ca 300 pixels in size. This is a bit marginal. While sufficent for most features, taxonomically important small features can get lost (bladed vs cylindrical spines, etc). Computation costs increase with image size but if possible I would suggest nearly doubling the image resolution for future work, particularly if the database used is going to be promoted as a standard for future contributions.

Response: We agree that some features cannot be unambiguously resolved using the 256x256 pixels image size. We did run some tests using bigger vignette images size, but on our server when images are larger than 320 px in size, the server crashes (even with 26 go of allocated RAM space). To test if bigger images can produce significantly better results, the database was reduced to about 13k images by removing some images of classes that contain more than 300 images. This way, we could increase the images size to 384 px without crash. However, results were not better, and in fact,

worse, as the overall accuracy drop to 87 %. This could be due to small images being resized, generating pixellisation that was learn as a feature and creating confusion between classes. Other testing were conducting by adding images to the database and resizing the dataset to 256 px to compensate for the increased RAM needed. Results were better than the original one presented in the first version of the manuscript were achieved, and are integrated in the revision of the manuscript to show as updated data as possible. Moreover, it has to be noted that small features that can be taxonomically important (such a bladed vs cylindrical spines) are likely to be difficultly learned by the network while the images quality, and intraspecific morphological variability would play a most important role, before the network can focus on such small features. We added in the manuscript lines 167- 170 a short discussion on this point.

Comment: The cited 17K image database is significantly smaller in numbers of species images. There are non radiolarian broad categories like 'spicule' - 8 categories, ca 8K images, plus supraspecific spp group categories - nearly 20 named,ca 4K images. The number of tagged species images is thus well under 10K. This distinction should be noted clearly in the ms.

Response: To address this issue, and as new images were generated since the original submission of this manuscript, the database was recently increased up to abut 22k images. Numbers were corrected in the text (lines 233 to 241), and database composition is now detailed. The database now contains 116 classes corresponding to species or groups of two to three species and containing 11,126 images, 7 classes corresponding to genera and containing 1,932 images, and 1 corresponding to family and containing 677 images. The 8 non-radiolarians classes contain 8,011 images.

Comment: For many taxa images have multiple image types, e.g. Acrosphaera spinosa specimens largely covered with some sort of milky bubble (preparation or image stacking artefact?) - 20 of 60 images. This problem is fairly common, seen in many folders. This particular image problem does not appear in routine sample preparation of similar materials and should be explained, as also the effect on identification accuracy.

Response: This issue is due to the glu's viscosity that might prevents air bubbles to escape from perforated type shells, common in collosphaeridae. Although not ideal, images of specimens containing bubbles, but still recognizable were kept into the database to integrate this variability into the neural network and prevent misidentification with class "bubble".

Comment: There are many, mostly minor issues with the taxonomic image database, tho some of these do not appear in the ms itself due to dropping rare image categories in the analysis.

Response: As the database is open-access and available online, it can be corrected when errors are reported. It was recently corrected for some images mis-identification, and classes names, also according to errors reporting below. We accept any suggestion to improve the taxonomical framework of the database.

Comment: There are really too many incorrect names being used in the ms and supporting image database. While not a problem for testing (the name tags as such have no effect on the system) these should be corrected. I have not gone through all 100 folders but based on a sampling of these I note the following. 'Strichocorys spp' - no such genus, content appears to be Phormostichoartus pitomorphus, Theocalyptra davisiana - correct name (since 30 years) is Cycladophora davisiana, Calyptra cervus instead of intended? name Corocalyptra cervus, tho recommended name is Eucecryphalus cervus - and see also below, Pseudodictyophimus gracilipes, kilmari, not Dictyophimus (again, since decades) - which importantly changes the family assignment.

Response: We acknowledge our mistakes : numerous names were corrected and new images and species were added to the database. The corrected names / taxonomy is now visible on the online database (autoradio.cerege.fr), on the downloadable file, and in the manuscript / figures. For instance, "Strichocorys spp" (error with an extra r) was corrected to "Phormostichoartus pitomorphus", Theocalyptra davisiana was

corrected to Cycladophora davisiana, Calyptra cervus was corrected to Corocalyptra cervus, Dictyophimus gracilipes to Pseudodictyophimus gracilipes, and other mistakes after a careful check on Lazarus et al., on WoRMS, and mikrotax.org, for a consistent and updated taxonomy.

Comment: There are also some typos such as Stylotractus instead of Stylatractus universus. Lastly there is at least one instance of possible oversplitting - i. e. Dictyocoryne trun- catum vs Euchitonia triangulum. What is the difference?

Response: Typos were corrected (all species of Stylatractus, Stichocorys, and others). With regards to oversplitting, Dictyocoryne truncatum was fused in the new CNN with E. triangulum, and even with D. profunda as both might be synonyms (as suggested by Boltovskoy) and exhibit a large inter and intra specific morphological variability (even more when specimens appear broken).

Comment: There are also a certain number of specimens in the individual taxon folders that are not con-specific. A brief sampling yields:

image name given correct name total images in category 14875 Cyrtopera langucula Artopilium undulatum 6

Response: the problematic image was moved to a new folder "Artopilium undulatum".

11233 Anthocyrtidium ophirense unknown but pores much too large to be conspecific 20

Response: This image was removed from the database as not identifiable.

01235 Zygocircus productus Zygocircus piscicaudatus 67

Response: As we see a lot of morphological gradation between both species in our specimens, the class is now called Zygocircus piscicaudatus productus and contain both species. A new class was created and contains Zygocircus capulosus specimens found in Miocene sample.

13329 Amphisphaera sp. B appears conspecific w. some specimens labeled as Drup- patractus irregularis 21/28

Response: The whole bi-spicular actinommidae group was updated and corrected.

mult. Calyptra cervus multiple Cycladophora species including [u1486-..] Cycladophora cabrilloensis

Response: A class Cycladophora cabrilloensis was added.

Comment: The large majority of the identifications (so far as the image quality allows) appear to be correct as monospecific classes, even if sometimes the name for the class are not. An attempt to provide a standard set of names (provisional of course as taxonomy is always being revised) was given by a group of taxonomists in Lazarus et al. 2015. I sugggest for the sake of data standardization that they use these, or at least provide a taxonomic appendix where they explain any variant usage.

Response: Classes names were corrected as a standardization effort by using Lazarus et al., 2015 and WoRMS.

Performance measurements

Comment: The time has come, the Walrus said, To talk of many things: Of shoes and ships and sealing-wax, Of cabbages and kings (Lewis Carroll, Through the Looking Glass, 1871). Radiolarians are to a highly unusual degree morphologically diverse. There are at least a dozen topologically highly distinct Baupläne alone in living/late Neogene assemblages. Distinctions between these broad morphologic groups are trivially easy. This is quite in contrast to most other clades of organisms, and in particular to planktonic foraminifera, which were used by this research group to initially develop their algorithms and work flows. Planktonic foraminifera are morphologically very conservative (once described to me by a foram specialist- as 'basically just popcorn'), and can be considered for an imaging system as a single broader category for analysis purposes. For radiolarians, it is really not very informative to know that the

system can distinguish between forms as radically different as, well, cabbages and kings. The true test of performance is its capacity to accurately identify and distinguish between species within topologic/taxonomically similar groupings, such as within radiolarian families/subfamilies. This is not only for purely taxonomic reasons, but also for the utitily of a system in applied research. Radiolarians encode ecologic-environmental information almost entirely at the level of individual species. Attempts to use genus or family level taxa as proxies in applied paleoenvironmental research have yielded almost no useful signals. Geologic age in the Cenozoic is partially recoverable at the genus level, but at a resolution so poor as to make it uninteresting for actual use.

Response: First of all, we have to consider that the neural network that we trained, and overall, the whole workflow is a compromise between distinguishing as many species as possible, and try to keep a good accuracy for each class which mostly depend on the growing number of images in each of them. The more images will be progressively added to the database, the more accurate will be the identification, and the more we will be able to go to the specific level.

Comment: The ms therefore needs to clearly separate the performance of the system in distinguishing between morphologically/taxonomically similar forms and the ability to distinguish extremely dissimilar forms. I suggest in both the statistical analyses of output and in the creation/ organisation of the figures, and the image database for download, that a clustering to higher categories is done, e.g. radiolarian orders and families, plus 'other' for non radiolarian categories such as particles, diatoms or background. The results for pairings of related taxa such as Lophophaena hispida and Peromelissa pha- lacra, both within the lophophaenid subfamily of Plagiacanthidae (performance values ca 70%) suggest that the accuracy for applied uses may be significantly lower than the current bulk statistics suggest. It is also important to report separately performance in identifying radiolarian species vs identifying broad categories such as 'spp.', diatoms, particles etc.

Response: To address this issue, and get a better idea of the system performance to

distinguish between similar, the accuracy was also computed within each family (see new Fig. 4). This accuracy is computed by taking the accuracy and number of test specimens for each class into account. We can see that this accuracy is about 91% for each family (e.g. 85% for the plagiacanthidae). To test the ability of the system to distinguish between dissimilar forms, the % of specimens correctly assigned to their family (vs specimens identified in a wrong family) was also computed and added to fig. 4. A short discussion was added lines 285 to 292. MoreRegarding the clustering to a higher categories, as suggested, Fig. 4 was emended to add the accuracy at the family level. The website for the online catalogue (autoradio.cerege.fr) also shows an order / family / genus / species organization. However, the downloadable database was left with the original ranking, as it is directly used for the training step, which cannot handle a complex ranking system (all class need to share the same rank level).

Comment: Lastly, when errors Are made, the nature of these is significant. It is hard to understand how this system could mis-identify a Pseudodictyophimus gracilipes [incorrectly named Dictyophimus gracilipes in the ms] with Hexacontium spp. - these taxa are in different orders, and have [to a taxonomic expert] fully different morphologies. Some sort of statistic giving not just the error rate, but the type of error - misclassified as to species in same family, different family in order, or different order should be given.

Response: We implemented in the training of the CNN (Marchant et al, in press) a workflow to extract the closest images based on the CNN vector loading, to identify likely misclassified images in the training data set. This is useful for checking bad manual identification, to groundchek the neural network training. However, as a figure is generated for each image tagged as " misclassified ", considering the size of the database (currently more than 21K images), the sum of all the generated figures is heavy and was not integrated in supplementary material. Although it might be confusing to understand why a P. gracilipes which is a nasselaria, might be confused with a Hexacontium spp, which is a spumellaria, it is usually not a question of how much are two species related, but how similar are 2 black and white tiny images showing

strange forms. From a far point of view, P. gracilipes is basically a somewhat rounded form exhibiting 3 to 4 spiny extension, depending on the point of view, which is also the case for Hexacontium spp when some spines are broken. Although not taxonomically closely related. If the P. gracilipes class only show 3 to 4 spined specimens, and Hexacontium 5 to 6 spined forms, depending on the view, a broken specimen of Hexacontium spp exhibiting 3 to 4 spines is likely to be classified in P. gracilipes. This is why including a large intraspecific morphological variability, with slightly broken specimens, bubble, and so on is important, as the majority of shells are not perfectly preserved in sediment.

Completeness of taxonomic/morphologic coverage

Comment: The authors have clearly made an effort to look at the entire assemblage of radiolarians, which is perhaps the most distinctive, laudable and novel aspect of this study. It must however be noted that the number of actual examined species is less than 80, while radiolarian diversity in tropics-subtropics is ca 500 (just in the sediments, not counting those, relatively few, species in the plankton that do not preserve).

Response: We add new images (about 5000) regarding rare species that were poorly images before, and new species not present in the samples previously investigated, were added to the database, as more samples were processed since the initial submission. The number of radiolarian classes is now up to 124 (with 101 classes with more than 10 images corresponding to the minimal number of images used in the training of the CNN) + 8 non radiolarian classes.

Comment: Thus only about 15% of the diversity in these assemblages has been incorporated into the study. Adding a larger percentage of the species is a stated goal of their project and quite correct. However, as more species are added, the number of closely related species pairings will increase, in relation to comparisons between distantly related forms. This is likely to have a negative effect on the performance of the system, as accuracy in similar pairings appears to be fairly low at present. This

may not affect using the system for classic assemblage based proxies of paleoenvironmental conditions as these can be based on relatively few, selected species or even species groups. There is not much demand however at the for this type of work, as it has been largely displaced by geochemical methods. Possibly having a cheap system to generate the data will revive it, but I am somewhat skeptical. Biostratigraphy however remains important, along with a variety of emerging themes related to evolution and biodiversity. These studies though need to use a larger fraction of the assemblage, distinguish closely related forms, and/or include rarer species. (Indeed it seems to be an instance of Murphy's Law that important biostratigraphic markers are so hard to find in many slides...). The usability of this system will only become apparent when it has grown to include more taxa, including many closely related and rare forms. The ms should make these limitations clear. There is no general answer to what level of accuracy is 'adequate', but I would suggest that for many biostratigraphic and biodiversity studies error rates should be closer to 1% than 10%, which may prove quite challenging for AI systems to achieve. Alternatively analytic methods in these fields will have to be revised to handle much noisier data.

Response: We acknowledge that our study does not encompass all the diversity of modern radiolarians, but is a proof of concept study based on a real test case. We are continuously trying to improve the system, so it can be used in a variety of studies, including its ability to distinguish very similar species for biostratigraphy or evolutionary studies. For example, more Pliocene and Miocene images were added to the database and which now exhibits new species of Dydimocyrtis and closely related Diartus species. In the original neural network from the first submission, Dydimocyrtis tetrathalamus tetrathalamus was recognized with an accuracy of 91%. In the new version of the neural network that was recently trained with more images, and corrected names, Dydimocyrtis tetrathalamus tetrathalamus is still recognized with an accuracy of 91%, while added species groups Dydimocyrtis antepenultima penultima is recognized with an accuracy of 96%, and Diartus hughesi petterssoni with 83%). More species are thus not likely to decrease the accuracy of the network, if enough images

are present in each class. In the same way, we are confident that with more and more images, classes composed of 2 or 3 species will be progressively divided in several distinct classes.

Comment: There is an additional issue in the adequacy of counting only a few hundred individuals to represent an assemblage of radiolarians. This is adequate only for the rather small fraction of species (usually <10% of the species richness) that are present at several percent abundance in the assemblage, as a closer reading of the short paper the authors cite (Fatela and Taborda) would reveal. While a cut-off of several percent is indeed frequently used in paleoenvironmental proxy studies, this is a seriously inadequate degree of coverage for either biostratigraphy or evolution/biodiversity research. Indeed, this problem is indirectly illustrated in the inadequate numbers of specimens for many species in their training-test image sets. There is in fact a very large body of sophisticated literature in ecology on determining the adequacy of sample sizes for different degrees of assemblage diversity and desired completeness of the resulting sample. See as starting points Chao et al. (2020) Ecol. Res.; Dornelas et al. (2012) Proc. R.Soc. B, and the brief discussion related to radiolarian assemblages in Lazarus et al. (2018) PeerJ. For radiolarians, in many cases the appropriate sample size is several thousand specimens.

Response: Depending on the goal and accuracy of the study, this issue can be easily addressed in the sample preparation by pouring a solution of the same sample in the 8 tanks of the decanter (or more or less tanks according to the abundance of radiolarian in the sediment). This way, no changes are required for the image acquisition part of the workflow. Several versions of the decanter for bigger cover slides (32x24 mm and 40x22 mm as seen in some papers) were added to the download platform, and then required a change in the acquisition part of the workflow (the acquisition software was develop so you can directly enter the number and size of cover slide used, so no change in the code is required). New versions of the decanter for other cover slide sizes can also be generated on demand. All versions are available at:

https://github.com/microfossil/Decanter

Sample coverage

Comment: Detailed information is given for this in the SOM, but essentially all material is from a single location in the western equatorial Pacific. I miss a discussion, or at least a disclaimer, of how geographic variation in morphology, or variation over time in lineages might affect the system's performance by blurring between species distinctions. It would also be nice to know in more detail the ages of the samples and the sources of the age information.

Response: More information was added about cores location. More material was added as more images were acquired since the initial submission. This material originates from other cores that cover a larger area in the WPWP. A discussion about variation in morphology and morphological and assemblages variation over time in lineages will be proposed in the following study as these parameters will be measured and are already observed in our data with consistent and good results (e.g. Dydimocyrtis tetrathalamus tetrathalamus with an accuracy of 91%, Dydimocyrtis antepenultima penultima with 96%, and Diartus hughesi petterssoni with 83%), again with the aim of increasing the accuracy and distinguishing radiolarians species group in groups of species, or genera, as more images are added to the database.

New sample preparation method

Comment: In this section a new variant of a coverslip holder is described. Although the goal of making slides with very few individuals seems to me to somewhat quixotic given the sample adequacy issues mentioned above, the idea of a custom designed holder that can be manufactured by 3-D printing is novel and is, adapted perhaps to full size cover slips, a useful addition to the literature. The chemical and other preparation steps are fairly standard and, though important to include, could be moved to the SOM.

Response: As mentioned above, new versions of the decanter of different sizes are

available at: https://github.com/microfossil/Decanter

Figures and Tables

Comment: The confusion matrix (fig. 4) is useful but a much more readable table listing each species, numbers/percents correctly and incorrectly classified and the top 3 error categories they were assigned to would be very helpful. I spent too much time scrolling figure 4 around my screen.

Response: As the confusion matrix is always used in deep learning studies, we would like to keep is if possible. We emended is to make is more easy to read, and add the family accuracy scores. To address the issue, the original excel file with a fixed class names column that enable a more efficient scrolling is provided as a supplementary material (Appendix B). As suggested, the % accuracy for each class, number of images in the test set, and top 3 error categories was also added to this supplementary material in a second excel file (Appendix C).

Comment: I think the citation to Lazarus et al. 2015 line 23 should be Lazarus 2005 while line 235 should be to Lazarus et al. 2015 - they have inverted these.

Response: Citations were corrected, thanks.

The original Figure 5 and associated paragraph were removed as they were originally used to show the accuracy of the network and the recall scores for each class, but we believe that the new Figure 5 (update of the original figure 6) and the updated confusion matrix (Figure 4) and new Appendix B and Appendix C are more usual and efficient ways to show the accuracy of the network and individual class scores.

Please also note the supplement to this comment:
https://cp.copernicus.org/preprints/cp-2020-76/cp-2020-76-AC1-supplement.zip
* * *
**Fig. 1.**

---

## Referee Comment (RC2) · Thore Friesenhagen (Referee) · 14 Aug 2020

**1. General Comments**

The development of an automated system for the collection of radiolarian census-data using a neuronal network is a consequent step to give over time-consuming work-flows to machines. Posting the codes as well as the organisation of a discretionary image-based radiolarian (training) dataset are a good practice, but it also means that maintaining the dataset will be one of the most important tasks for the future. The manuscript is well done and requires only minor revisions. The following annotations and questions should be considered and/or answered in the final publication.

[Figure]

**2. Scientific Questions**

**2.1. Convolutional Neuronal Network**

I miss a short introduction about neuronal networks and the "k-nearest neighbours" algorithm for readers who are not familiar with these terms.

**2.2. CNN Database**

As mentioned in the script's introduction, one and the same specimen may be referred to different species/classes depending on the experience, subjective interpretation and/or taxonomic "education" of the researcher (e.g. Fenton et al. (2018) for planktonic foraminifera). Thus, to reduce the number of possibly mistakenly identified specimens in the training dataset, having at least one more taxonomic expert checking the correctness of the species determination of specimens within the dataset could increase the reliability of the dataset.

**2.3. Image Stacking**

Does the transparency of the radiolarian shells produce any problems for the image stacking, especially in case of smaller and more delicate specimens? Figure 3j) shows a specimen of the species Collosphaera tuberosa. Its contours are diffuse. Is this a common "problem" for this species? Does this affect the identification accuracy for this species and may be one reason for the relatively high value of confusion with Solenosphaera zanguebarica?

**2.4. Early Ontogenetic Stages/Juveniles**

The collection of census data for planktonic foraminifera avoid juvenile specimens (e.g. Davis et al. (2019) only investigated the >125$\mu$m fraction), because their identification is often very difficult (Fenton et al., 2018). Is there a lower size limitation for radiolarian specimens to be detected and identified by the new system? Is the system able to distinguish between early ontogenetic stages and broken specimens? Does the size of specimens affect the accuracy of the automated species determination?

**2.5. Image Acquisition**

What is the procedure for (intact) specimens which extend over the borders of the 324 FOV and are parted/bisected? Is the program able to identify these specimens as being intact? In this case, are these specimens prevented from being "double-counted" by the system?

**2.6. Influence of Orientation**

Closely related species tend to show a similar morphology and are often only distinguishable by details. Since the sample preparation bases on random settling, the orientation of a single specimens may not be ideal to enable the program to recognise these morphological details. What is the procedure for specimens which do not show an ideal orientation for determination?

**2.7. Morphological Measurements**

I give the authors credit for implementing morphometric measurements. In combination with census data they may provide additional and valuable information for palaeoenvironmental reconstructions and evolutionary studies. Although this paper clearly focuses on the collection of census data, the accuracy of the morphometric measurements should be given as well. To what extend do differences in specimen orientation affect the accuracy and intraspecific comparability?

**3. Technical Corrections**

**3.1. Text**

-l. 62-63: The sentence contains two times the phrase "promising results".

-l. 86: A comma is missing after "6.3ka". "[. . .] 3-4cm, (6.3ka[,] de Garidel-Thoron et al., 2005) [. . .]"

-l. 273: there is a closing bracket at the end of the sentence, but I could not figure out the corresponding, opening counterpart. "[. . .]and that 150 images represent the

original dataset for this class; Fig. 5 green square). [. . .]"

-l. 359: The semicolon may be replaced by a closing bracket. "[. . .] palaeoenvironmental proxies such as SSTs (e.g., radiolarian-based palaeotemperatures for [. . .], Kamikuri, 2017;])] and paleoproductivity [. . .]

3.2. Figures

-Fig. 2: A space is missing in the text for step 7. "7.[ ]Identification of every single particle using a trained CNN."

-Fig. 4: The printed version is difficult to read, because the font size of the species names is relatively small. The digital figure requires a lot of scrolling.

-Fig. 5: Several names of species overlap and make it impossible to read them.

-Fig. 6 e,f: The percentage numbers are difficult to read, because they overlap with black bars within the figure.

4. References

Davis, Catherine V., et al. "Seasonal and interannual changes in planktic foraminiferal fluxes and species composition in Guaymas Basin, Gulf of California." Marine Micropaleontology 149 (2019): 75-88.

Fenton, Isabel S., et al. "Factors affecting consistency and accuracy in identifying modern macroperforate planktonic foraminifera." Journal of Micropalaeontology 37.2 (2018): 431-443.

---

## Referee Comment (RC3) · Anonymous Referee #3 · 19 Aug 2020

Structure/composition of the Paper

While the paper looks overwhelming due to the number of pages, the content is actually concise. The information written there is not too long nor short. The structure of the paper follows the usual format (introduction, methodology, results and discussion).

Methodology/approach to the Problem

Their workflow is a whole and complete system, which starts at image acquisition and ends at classification. The workflow hopes to ease the tedious task of identifying specimen from samples, which requires extensive and consistent taxonomic knowledge of the observer to correctly identifyÂăradiolarians. Research was done well, as it can be

seen that they have explained the steps in great detail (including the measurements).

As for the AI specific topic, they have used a usual Deep Learning approach. They have used ResNet and finetune it on their dataset. They have also included non-Radiolarian classes, which I believe provided an edge especially since they will be classifying things straight from the image acquisition (w/o humans to remove the non-Radiolarian particles). They have acquired a lot of samples, so they did not struggle that much on this part. Overall what I can see here is that the acquisition and segmentation of images are more tedious than the actual training and classification of Radiolarians.

For the purpose of training the Convolutional Neural Network (CNN) for identification ofÂăRadiolarians, they developed and released AutoRadio (AutomatedÂăRadiolarian). To encourage participation and contributions on adding more images to AutoRadio, they provided a very detailed protocol to standardize the way of obtaining images. Even the file for 3D printing Decanter, used to prepare the slides, is provided for everyone to use. The repository for Decanter also includes a video for the modified random settling protocol.

It is suggested that a section briefly discussing the convolutional neural network model should have been included. The approach fundamentally relies on the model and hence it is necessary to detail how it is applied so as to properly justify the solution for automated identification. As such, the section shall essentially include the following: CNN overview, model architecture, and training approach (transfer learning, loss, etc.)

A minor concern is that I noticed that the Random Settling Protocol, as discussed starting in line 95 and the video (https://www.youtube.com/watch?v=veRmKI4rGTo) differ in the series of steps taken for the preparation of theÂăradiolarianÂăslides. I recognize that some steps are possibly not filmed for brevity, and the difference in steps might suggest that what is written on paper may not be strictly followed. But the motivated reader who wishes to contribute and follow the protocol may feel confused at first. I also noticed that in the video, the sample taken only amounted to 0.1 mg, but in the

protocol the recommended amount is 0.6 mg (line 130, step #15), as it corresponds to the best compromise to ensure that a sufficient number of radiolarian specimens are covered and at the same time the specimens are not crowded and not touching one another, as discussed in subsequent sections that overlaps might affect the ability of the workflow to identify radiolarians. What I thought is that in cases where the amount of samples is limited, taking at most 0.6 mg would be enough. All things taken, the inclusion of the video is very helpful.

Another concern is about imbalance in classes, which is actually common among Radiolarian studies. Reading on the documentation of ParticleTrieur, the recommended number of images per class is 50 at minimum and preferably at least 200 images per class, which can be very difficult to achieve especially on rare radiolarian species. Commonly, data augmentation is performed to address the issue of class imbalance. But augmenting the data has to ensure that variations applied to the image still preserve the class/label after applying transformations. Hence, careful application of augmenting data must be ensured. ParticleTrieur also makes use of weighted loss functions, which is another good way of handling class imbalance.

I agree that ideally, adding more data on rare species would improve the trained model so paper emphasizing the possibility of collaboration through adding more images to AutoRadio and detailing on how one can contribute is really a good step.

Discussion of the Results

They have used the usual metrics (Accuracy, Precision, Recall, Confusion Matrix). The results were good since image acquisition and segmentation methodology is already profound, their data is quite large ($\sim$17k samples total), and they have reported an overall accuracy of 90%.

---

## Author Comment (AC2) · 26 Aug 2020

Anonymous referee.

Comment: Structure/composition of the Paper: While the paper looks overwhelming due to the number of pages, the content is actually concise. The information written there is not too long nor short. The structure of the paper follows the usual format

(introduction, methodology, results and discussion).

Response: Thank you very much for your review. Indeed, lots of things are discussed in the manuscript, and it is difficult to reduce it too much, but we try to keep things concise.

Comment: Methodology/approach to the Problem: Their workflow is a whole and complete system, which starts at image acquisition and ends at classification. The workflow hopes to ease the tedious task of identifying specimen from samples, which requires extensive and consistent taxonomic knowledge of the observer to correctly identify radiolarians. Research was done well, as it can be seen that they have explained the steps in great detail (including the measurements).

Response: We aimed at developing a complete system for radiolarian research by using the expertise developed at CEREGE for several decades now. Everything was detailed as much as possible to enable scientists from other laboratories to use this method for their research.

Comment: As for the AI specific topic, they have used a usual Deep Learning approach. They have used ResNet and finetune it on their dataset. They have also included non-Radiolarian classes, which I believe provided an edge especially since they will be classifying things straight from the image acquisition (w/o humans to remove the non-Radiolarian particles). They have acquired a lot of samples, so they did not struggle that much on this part. Overall what I can see here is that the acquisition and segmentation of images are more tedious than the actual training and classification of Radiolarians.

Response: Indeed, non radiolarian classes were used to enable the system to identify the sediment material straight from the vial. Acquisition and segmentation parts were indeed the more tedious parts due to the complex morphology and composition of radiolarians shells.

Comment: For the purpose of training the Convolutional Neural Network (CNN) for identification of Radiolarians, they developed and released AutoRadio (Automated Radiolarian). To encourage participation and contributions on adding more images to AutoRadio, they provided a very detailed protocol to standardize the way of obtaining images. Even the file for 3D printing Decanter, used to prepare the slides, is provided for everyone to use. The repository for Decanter also includes a video for the modified random settling protocol.

Response: We tried to make this whole new protocol as accessible as possible for future radiolarian studies.

Comment: It is suggested that a section briefly discussing the convolutional neural network model should have been included. The approach fundamentally relies on the model and hence it is necessary to detail how it is applied so as to properly justify the solution for automated identification. As such, the section shall essentially include the following: CNN overview, model architecture, and training approach (transfer learning, loss, etc.).

Response: A small discussion was added as suggested by reviewer 1 and anonymous reviewer. More details about the used CNN are provided.

Comment: A minor concern is that I noticed that the Random Settling Protocol, as discussed starting in line 95 and the video (https://www.youtube.com/watch?v=veRmKI4rGTo) differ in the series of steps taken for the preparation of the radiolarian slides. I recognize that some steps are possibly not filmed for brevity, and the difference in steps might suggest that what is written on paper may not be strictly followed. But the motivated reader who wishes to contribute and follow the protocol may feel confused at first. I also noticed that in the video, the sample taken only amounted to 0.1 mg, but in the protocol the recommended amount is 0.6 mg (line 130, step #15), as it corresponds to the best compromise to ensure that a sufficient number of radiolarian specimens are covered

and at the same time the specimens are not crowded and not touching one another, as discussed in subsequent sections that overlaps might affect the ability of the workflow to identify radiolarians. What I thought is that in cases where the amount of samples is limited, taking at most 0.6 mg would be enough. All things taken, the inclusion of the video is very helpful.

Response: The video shows how to use the decanter. The part were samples are chemically prepared is thus not included in the video. The preparation protocol was very slightly emended since the original publication and now matches the protocol that was actually used to prepare 400 samples for the next study and that is visible in the video. Regarding the amount of sample taken in the video (0.1 mg), it was an annotation mistake that was corrected. It is indeed recommended to use between 0.6 and 1.0 (not 0.1) mg of material. This was corrected in the text and in the video.

Comment: Another concern is about imbalance in classes, which is actually common among Radiolarian studies. Reading on the documentation of ParticleTrieur, the recommended number of images per class is 50 at minimum and preferably at least 200 images per class, which can be very difficult to achieve especially on rare radiolarian species. Commonly, data augmentation is performed to address the issue of class imbalance. But augmenting the data has to ensure that variations applied to the image still preserve the class/label after applying transformations. Hence, careful application of augmenting data must be ensured. ParticleTrieur also makes use of weighted loss functions, which is another good way of handling class imbalance.

Response: Indeed, some radiolarian species can be very rare to tricky to found. 200 specimens are recommended per class, 50 is a minimum for accurate results, and here we decided to use classes with at least 10 specimens to train as many classes as possible where more images will be progressively added. This even if these classes are not very accurate, the system can start to recognize them and already help with the identification. The presence of classes with few images do not decrease the accuracy of the overall network and do not affect the other classes.

Comment: I agree that ideally, adding more data on rare species would improve the trained model so paper emphasizing the possibility of collaboration through adding more images to AutoRadio and detailing on how one can contribute is really a good step.

Response: More images will be progressively added to the database as we process samples, to cover the rare and under-represented species. We also encourage people to send us their own pictures to cover the morphological variability that is not due to actual change in the shape of specimens, but to different acquisition settings, material, and so on. With active collaboration, the database should be expanding quickly.

Comment: Discussion of the Results: They have used the usual metrics (Accuracy, Precision, Recall, Confusion Matrix). The results were good since image acquisition and segmentation methodology is already profound, their data is quite large (17k samples total), and they have reported an overall accuracy of 90%.

Response: Again, we thank the reviewer for its comments about the MS.

---

## Author Comment (AC3) · 26 Aug 2020

Thore Friesenhagen (Referee)

thore.friesenhagen@unibas.ch

General comments

Comment: The development of an automated system for the collection of radiolarian census-data using a neuronal network is a consequent step to give over time-

consuming workflows to machines. Posting the codes as well as the organisation of a discretionary image-based radiolarian (training) dataset are a good practice, but it also means that maintaining the dataset will be one of the most important tasks for the future. The manuscript is well done and requires only minor revisions. The following annotations and questions should be considered and/or answered in the final publication.

Response: We thank the reviewer for its comments about the MS, a detailed answer to each of its raised point is given below.

Comment: I miss a short introduction about neuronal networks and the "k-nearest neighbors" algorithm for readers who are not familiar with these terms.

Response: A short sentence was added about the neural network that was used, however, we refer to the very detailed paper of Marchant et al. (Accepted) for more information about accurate description of the algorithm.

Comment: As mentioned in the script's introduction, one and the same specimen may be referred to different species/classes depending on the experience, subjective interpretation and/or taxonomic "education" of the researcher (e.g. Fenton et al. (2018) for planktonic foraminifera). Thus, to reduce the number of possibly mistakenly identified specimens in the training dataset, having at least one more taxonomic expert checking the correctness of the species determination of specimens within the dataset could increase the reliability of the dataset.

Response: Overall, three expert reviewed the database: Martin Tetard, based on numerous and consistent publications, Giuseppe Cortese a radiolarian taxonomist expert, and the first reviewer, David Lazarus, that points out numerous correction on the database.

Comment: Does the transparency of the radiolarian shells produce any problems for the image stacking, especially in case of smaller and more delicate specimens? Figure

3j) shows a specimen of the species Collosphaera tuberosa. Its contours are diffuse. Is this a common "problem" for this species? Does this affect the identification accuracy for this species and may be one reason for the relatively high value of confusion with Solenosphaera zanguebarica?

Response: As the shells are outlined by the different refractive indices between the shell and the mounting medium, we don't experience any issue with the stacking step, even with small and / or delicate shells. Lots of time was spent for finding the best parameters with regard to the stacking method on Helicon Focus. The diffuse contours of the specimen of Collosphaera tuberosa is due to its position on the slide. This specimen was slightly out of focus from the 150 $\mu$m range imaged using the automated stacking technique. However, this specimens is the best preserved we have for now, explaining why it was chosen for a plate. This problem is not common but on some slides, it may happen that some specimens are out of focus from the stacking range. This does not usually affect the identification. Most of the confusion between C. tuberosa and S. zanguebarica was due to the few number of specimens that was imaged in both classes. Now, with more images, these two classes are no longer confused with each other in the new version of the CNN.

Comment: The collection of census data for planktonic foraminifera avoid juvenile specimens (e.g. Davis et al. (2019) only investigated the >125$\mu$m fraction), because their identification is often very difficult (Fenton et al., 2018). Is there a lower size limitation for radiolarian specimens to be detected and identified by the new system? Is the system able to distinguish between early ontogenetic stages and broken specimens? Does the size of specimens affect the accuracy of the automated species determination?

Response: We only work on the fraction > 50 $\mu$m to avoid lots of juvenile specimens that are often difficult to identify as early features are shared between numerous species, and lots of broken shells. However, the ontogenetic stages are visible in numerous classes and, when they are sufficiently imaged, can be distinguished. The size

of specimens do not seems to affect the accuracy of the system as most of the images are above the images size (256 px) used to train the CNN.

Comment: What is the procedure for (intact) specimens which extend over the borders of the 324 FOV and are parted/bisected? Is the program able to identify these specimens as being intact? In this case, are these specimens prevented from being "double-counted" by the system?

Response: For specimens that are "cut" between two FOVs, if a part of the shell contains the first chambers (usually for nasselaria) it should be identified as the correct class, and the second part should be identified as "broken", to prevent a double identification in the correct class.

Comment: Closely related species tend to show a similar morphology and are often only distinguishable by details. Since the sample preparation bases on random settling, the orientation of a single specimens may not be ideal to enable the program to recognise these morphological details. What is the procedure for specimens which do not show an ideal orientation for determination?

Response: The intra-specific morphological variability generated by the orientation of specimens in images is mostly covered by the number of images present in each class. However, most of the specimens settled the same way for each specimen of a class. Also, if a specimen is oriented in a way that its identification is not possible (e.g. the under view of a nasselaria, which is very rare), it is identified as "broken".

Comment: I give the authors credit for implementing morphometric measurements. In combination with census data they may provide additional and valuable information for palaeoenvironmental reconstructions and evolutionary studies. Although this paper clearly focuses on the collection of census data, the accuracy of the morphometric measurements should be given as well. To what extend do differences in specimen orientation affect the accuracy and intraspecific comparability?

Response: The morphometric measurements of every specimens, averaged for each class, will be presented for two cores in a next study that is in preparation. As the morphometric measurements are performed on the outline of each specimens, we are confident regarding their accuracy. Regarding the bias generated by the specimens' orientation, as specimens usually fall on the same way for every specimen of each class, we are confident that most of the observed variability is due to actual change in shape, and not to change in their orientation between samples.

Comment: -l. 62-63: The sentence contains two times the phrase "promising results". -l. 86: A comma is missing after "6.3ka". "[. . .] 3-4cm, (6.3ka[,] de Garidel-Thoron et al., 2005) [. . .]" -l. 273: there is a closing bracket at the end of the sentence, but I could not figure out the corresponding, opening counterpart. "[. . .]and that 150 images represent the C3 original dataset for this class; Fig. 5 green square). [. . .]" -l. 359: The semicolon may be replaced by a closing bracket. "[. . .] palaeoenvironmental proxies such as SSTs (e.g., radiolarian-based palaeotemperatures for [. . .], Kamikuri, 2017;[)]) and paleoproductivity [. . .] -Fig. 2: A space is missing in the text for step 7. "7.[ ]Identification of every single particle using a trained CNN." -Fig. 4: The printed version is difficult to read, because the font size of the species names is relatively small. The digital figure requires a lot of scrolling. -Fig. 5: Several names of species overlap and make it impossible to read them. -Fig. 6 e,f: The percentage numbers are difficult to read, because they overlap with black bars within the figure.

Response: -l. 62-63: The was corrected and the double word was removed. -l. 86: The comma was added. -l. 273: The closing bracket was removed. -l. 359: The semicolon was replaced by a closing bracket. -Fig. 2: A space was added. -Fig. 4: This figure was revised and the species is now slightly bigger. To help with the reading of this figure, an excel spreadsheet was added as a supplementary material, with a fixed first column (classes name) that help with the scrolling. -Fig. 5: This figure was removed as discussed in the reviewer 1's response to comment. -Fig. 6 e,f: This was corrected.

Thank you for all your comments.

---

## Editor Decision (ED1)

Dear Authors,

I carefully read your responses to the reviewer's comments and your revised manuscript. You addressed most of the reviewer's comments and you improved the manuscript. Thank you.

However, there are a few points that I would like to you to work on. Most of them are related to open questions of the reviewers in their comments. While you answer very well to these questions in your point to point response, too often, you do not include that information in the text of your manuscript. I think you should include such information because, if the reviewer asked that question, it is most probably because the information was missing or should be clarified in the manuscript. Here is the list of things I would like you to consider in a revised version of your manuscript.

- I suggest that you add the explanation provided to Lazarus comment's about the glue somewhere in the text, maybe in section 2.3
- I also suggest to add on line 286 : "(…) and to send any suggestion to improve the taxonomical framework of the database" or something similar.
- I suggest you add something in the discussion about the usefulness of improving the identification to a level that is useful for paleoenvironmental research.
- While your answer to Lazarus comment at bottom of your page 5 (your point by point response to reviewers), I still think you should add a comment about the usability of this system and his limitation in the text of the MS.
- Again, it might be interesting to add a comment in the text about the appropriate sample size for radiolarian in terms of number of specimens to be counted.
- Sample coverage: I suggest that you add a disclaimer about how geographic variation in morphology, or variation over time in lineages might affect the system's performance by blurring between species distinctions. I understand that you are going to detail this in an upcoming study, but I also acknowledge that the reviewer has a good point here.
- Reviewer Lazarus suggested to move some of the preparation method to SOM. Why do you think it is useful to keep it in the main text body? Maybe you have very good reasons to keep here, but please provide an answer.
- It would be nice to mention in the text of the MS that autostacking does not affect correct identification, even for delicate specimens such as the one mentioned by reviewer Friesenhagen (something more than point 3 of your conclusion).
- Once again, you answer the question of reviewer Friesenhagen about the ontogenic stages, but you do not add anything about it in the MS. I invite you to explain somewhere that ontogenetic stages are visible in numerous classes and, when they are sufficiently imaged, can be distinguished, for instance in section 2.3
- Friesenhagen asks: "What is the procedure for (intact) specimens which extend over the borders of the 324 FOV and are parted/bisected? (…)". You should include your answer in the manuscript.
- Fig.1 Please add a scale.
- Fig. 2 There is some shading on the legend of each panel that looks unnecessary to me, making the letters less clear.
- L 213: What is a ResNet50? please add a reference to resnet50 topology.

- L 277 and following: I guess it should read "(…) then trained to be recognised with a current overall  accuracy of just above 90 % (90.1 %) over every class. The average precision is above 85 % (85.6 %) and the (…)".
- Fig 4. Caption: add that the figure is also available as an Excel spreadsheet in supplement.
- L 305 and following: Fig. 6 should be changed to Fig. 5
- L 485. Marchant at al should be accepted, or in press, but please make it consistent throughout the MS.
- Section Biostratigraphy: I'm puzzled that no data or figure supports this section. Where is the Excel sheet that you are referring too?
- Figure A1 is not called for in the text. The caption of this figure should contain some information about the samples
- I suggest you also refer to your web site and make more publicity about its content. It is quite nice!
- L 395 (competing interest) : I think you should declare that one co-author, Luc Beaufort, is an associate editor of Climate of the Past.

I think these points can be very easily addressed, and I'm therefore looking forward to your revised manuscript.

Best Regards

Pierre Francus, associate editor